# Altered Conformational Landscape upon Sensing Guanine Nucleotides in a Disease Mutant of Elongation Factor-like 1 (EFL1) GTPase

**DOI:** 10.3390/biom12081141

**Published:** 2022-08-19

**Authors:** Jesús Pérez-Juárez, Juana Virginia Tapia-Vieyra, Gabriel Gutiérrez-Magdaleno, Nuria Sánchez-Puig

**Affiliations:** 1Departamento de Química de Biomacromoléculas, Instituto de Química, Universidad Nacional Autónoma de México, Circuito Exterior s/n, Ciudad Universitaria, Ciudad de Mexico 04510, Mexico; 2División de Ciencias Naturales e Ingeniería, Universidad Autónoma Metropolitana, Unidad Cuajimalpan Avenida Vasco de Quiroga 4871, Ciudad de Mexico 05348, Mexico

**Keywords:** conformational change, EFL1, guanine nucleotides, ITC, magnesium ions, SBDS, SDS

## Abstract

The final maturation step of the 60S ribosomal subunit requires the release of eukaryotic translation initiation factor 6 (human eIF6, yeast Tif6) to enter the pool of mature ribosomes capable of engaging in translation. This process is mediated by the concerted action of the Elongation Factor-like 1 (human EFL1, yeast Efl1) GTPase and its effector, the Shwachman-Bodian-Diamond syndrome protein (human SBDS, yeast Sdo1). Mutations in these proteins prevent the release of eIF6 and cause a disease known as Shwachman–Diamond Syndrome (SDS). While some mutations in EFL1 or SBDS result in insufficient proteins to meet the cell production of mature large ribosomal subunits, others do not affect the expression levels with unclear molecular defects. We studied the functional consequences of one such mutation using *Saccharomyces cerevisiae* Efl1 R1086Q, equivalent to human EFL1 R1095Q described in SDS patients. We characterised the enzyme kinetics and energetic basis outlining the recognition of this mutant to guanine nucleotides and Sdo1, and their interplay in solution. From our data, we propose a model where the conformational change in Efl1 depends on a long-distance network of interactions that are disrupted in mutant R1086Q, whereby Sdo1 and the guanine nucleotides no longer elicit the conformational changes previously described in the wild-type protein. These findings point to the molecular malfunction of an EFL1 mutant and its possible impact on SDS pathology.

## 1. Introduction

Ribosomes are the cellular nanomachines that synthesise proteins in all organisms. Eukaryotic ribosomes are formed by the association of the small 40S and the large 60S subunits in a highly regulated process that requires the participation of the three RNA polymerases and several small nucleolar ribonucleoproteins (snoRNPs) and assembly factors (details have been reviewed elsewhere [1,2]). The last cytoplasmic maturation step of the nascent 60S subunit couples the release of anti-association factor eIF6 (yeast Tif6) with a quality control assessment of the P-site and the GTPase-associated centre via the joint action of the Elongation Factor-like 1 (EFL1) GTPase and the Shwachman-Bodian-Diamond Syndrome (SBDS) protein [3,4]. EFL1 is a molecular motor whose conformational plasticity is central for the release of eIF6 and is finely regulated by interactions with SBDS, the 60S subunit, and guanine nucleotides. SBDS acts as a GDP exchange factor or GEF for EFL1, favouring the active GTP-bound conformation of the enzyme [5,6]. Based on low-resolution cryo-EM data, EFL1 binds the GTPase-associated centre of the 60S subunit and undergoes an SBDS-dependent conformational change, whereby its catalytic module (domains I and II) sterically displaces eIF6 [7]. In this conformation, domain I of EFL1 is now anchored in the sarcin–ricin loop (SRL) which activates GTP hydrolysis by driving EFL1 into a GDP-bound conformation that in turn promotes its dissociation and that of SBDS [7]. In humans, defects in the production and function of ribosomes lead to a group of diseases referred to as ribosomopathies [8,9,10]. The Shwachman–Diamond Syndrome (SDS; OMIM #260400; #617941) is one such disorder associated with bone marrow failure and a high risk of developing myelodysplastic syndrome and acute myeloid leukaemia, together with exocrine pancreatic dysfunction, anaemia, neutropenia, and skeletal abnormalities [11,12]. SDS is an autosomal-recessive disease caused in most cases by mutations in the *SBDS* gene [13] and to a lesser extent by mutations in the gene encoding GTPase *EFL1*. Mutations in any of these proteins prevent the release of eIF6 from the 60S subunit, with the consequent imbalance of mature 60S subunits entering the pool of actively translating 80S ribosomes and a decrease in global translation [14,15].

The most common mutations in SBDS result in protein haploinsufficiency either via the disruption of the donor splice site of intron 2 or rearrangements that prematurely truncate the protein. Additionally, missense mutations that destabilise the protein fold or alter surface epitopes have also been reported, and their defect in a ribosome-free context is to weaken the interaction with EFL1 [13,16]. Different mutations have also been reported for EFL1 [15,17,18], although the molecular significance of those that do not affect the protein expression levels is still unclear. An initial approach using molecular dynamics simulations of three such mutants (R1095Q, M882K, and T127A) suggests they adopt a different conformation from the wild type [19]. Similarly, Small Angle X-ray Scattering (SAXS) studies on the yeast ortholog of mutant EFL1 R1095Q have revealed differences in the flexibility of the protein [17].

Diverse mosaic somatic genetic events have been described in the hematopoietic cells of SDS patients that bypass or compensate for the defects in ribosome production caused by germline *SBDS* mutations. In the bone marrow of patients, the most frequent germline alteration is an isochromosome for the long arms of chromosome 7, resulting in a double dose of the *SBDS* mutation that disrupts the donor splicing site but allows the production of a small amount of normal SBDS protein [20,21]. Interstitial deletions in the long arms of chromosome 20 rescue the deficiency of mature ribosomes by decreasing the dosage of eIF6, whose gene is lost in this chromosome deletion [22,23]. Other somatic genetic events that compensate for SBDS deficiency in SDS result in frameshift mutations leading to *EIF6* haploinsufficiency or missense mutations in conserved residues that destabilize the eIF6 protein or disrupt its binding interface with the 60S subunit [24,25]. This alleviating mechanism seems evolutionarily conserved, since yeast cells lacking *SDO1*, the *SBDS* ortholog, develop equivalent suppressor mutations in *TIF6* that attenuate the slow-growth phenotype and the subunit-joining defect [4].

Previous reports have shown that the expression levels and protein fold of *S. cerevisiae* Efl1 R1086Q, equivalent to mutation R1095Q described in SDS patients, are comparable to that of the wild-type protein [17]. So, the detrimental effect may not relate to a lack of enough protein to satisfy the physiological needs of the cell. Consistent with this, here, we show that mutant Efl1 R1086Q cannot efficiently release Tif6 from the 60S subunit using an ex vivo release assay with purified components. To gain further insights into the malfunction of EFL1 mutations and possible alterations in their communication with SBDS and the guanine nucleotides, we characterised the binding energetics of *S. cerevisiae* Efl1 R1086Q using isothermal titration calorimetry. The data presented here suggest this mutant transition through a complex conformational landscape upon binding to Sdo1 and guanine nucleotides distinct from that previously described for wild-type Efl1. The interaction with MgGTP did not induce a conformational change as observed for other GTPases and native Efl1; instead, binding occurred via rigid-body interaction. Similarly, Sdo1 could not elicit the structural and energetic effects expected to function as its guanine exchange factor. It no longer stabilised the GTP-bound conformation but favoured the complex with GDP. These alterations could be partially alleviated by a second variant located far away from domain IV, where the R1086Q mutation locates, suggesting that an intramolecular network communicates the effector module of Efl1 with the catalytic one. Considering that this mutant cannot efficiently release Tif6 from the 60S ribosomal subunit, it is plausible that these alterations transmit into the ribosomal context, decreasing the number of mature subunits produced.

## 2. Materials and Methods

### 2.1. Protein Expression and Purification

*S. cerevisiae* Efl1 wild type and mutants R1086Q, L910K, T33A, and R1086Q P151L were recombinantly expressed in *S. cerevisiae* and purified as described elsewhere [6,17,26]. Protein purity was assessed with SDS-PAGE, and the samples were stored in 50 mM HEPES-KOH pH 8.0, 150 mM NaCl, 5 mM MgCl_2_, and 10% glycerol at −80 °C until further use. Yeast SBDS (Sdo1) was recombinantly expressed in *Escherichia coli* C41 and purified as described in [26,27].

### 2.2. Purification of S. cerevisiae 60S Ribosomal Subunits

Ribosomal 60S subunits were purified from *S. cerevisiae* strain JD1370 (*MATa trp1 ura3 leu2 pep4::HIS3 nuc1::LEU2*) [28] as described in [29] with some modifications. Yeast cells were grown to an OD_600_ of 0.8–1.0 in 1 L of YEPD medium. Prior to collection, the cells were incubated for 2 min with 100 µg/mL cycloheximide to stop protein synthesis and chilled at 4 °C for further 15 min. Cells were harvested by means of centrifugation and washed twice with cold lysis buffer (100 mM potassium acetate, 20 mM HEPES-KOH (pH 7.4), 10.5 mM magnesium acetate, 1 mg/mL heparin, 2 mM DTT, 0.1 mM PMSF, 0.1 mM benzamidine, and 0.5 mM EDTA). Cells were resuspended in the same buffer and disrupted by means of friction with glass beads in a FastPrep-24 homogenizer (MP Biomedicals, Irvine, CA, USA) with four rounds of 30 s followed by 5 min rest on ice. The resulting lysate was cleared using centrifugation at 11,768× *g* at 4 °C for 20 min. The supernatant was placed on top of a 2.5 mL sucrose cushion (100 mM potassium acetate, 20 mM HEPES-KOH (pH 7.4), 10.5 mM magnesium acetate, 1 M sucrose, 500 mM potassium chloride, 0.5 mM EDTA, 2 mM DTT, 0.1 mM PMSF, and 0.1 mM benzamidine) and centrifuged for 2.5 h at 222,529× *g* in a 70Ti rotor (Beckman Coulter, Brea, CA, USA). Following centrifugation, the pellet was resuspended in high-salt wash buffer consisting of lysis buffer supplemented with 1.5 M potassium chloride. Insoluble material was removed using centrifugation, and soluble material was layered on top of a 2.5 mL sucrose cushion and centrifuged for 2.5 h at 222,529× *g* as mentioned above. The pellet was resuspended in subunit separation buffer (50 mM HEPES-KOH (pH 7.4), 10 µM MgCl_2_, 500 mM potassium chloride, 2 mM DTT, 0.1 mM PMSF, and 0.1 mM benzamidine) and stirred on ice for 1 h to dissociate the ribosomal subunits. In total, 50–80 optical units measured at 254 nm were loaded on top of a 10–40% linear sucrose gradient (50 mM HEPES-KOH (pH 7.4), 10 µM MgCl_2_, 500 mM potassium chloride, 2 mM DTT, 0.1 mM PMSF, and 0.1 mM benzamidine) and centrifuged for 6 h at 70,358× *g* in a 45Ti rotor (Beckman Coulter, Brea, CA, USA) at 4 °C. The gradient was fractionated by monitoring the absorbance at 254 nm. The 60S subunits were pooled together and pelleted at 222,529× *g* for 3 h in a 70Ti rotor. Finally, 60S subunits were resuspended in storage buffer (20 mM HEPES-KOH (pH 7.4), 100 mM potassium acetate, 10.5 mM magnesium acetate, 2 mM DTT, 0.1 mM PMSF, 0.1 mM benzamidine, and 250 mM sucrose) and stored at −80 °C until further use. The concentration of purified 60S ribosome subunits was determined by measuring the absorbance at 260 nm and using an extinction coefficient of 4 × 10^7^ cm^−1^ M^−1^ [30].

### 2.3. Tif6 Release Assay

The release of Tif6 (yeast orthologue of eIF6) was performed as previously described in [3,15]. Reaction mixtures contained 10 nM 60S subunits, 40 nM Tif6, 50 nM Efl1 variant, 50 nM Sdo1, and 100 µM GTP in buffer of 10 mM Tris-HCl at pH 7.4, 75 mM KCl, and 1.5 mM MgCl_2_ supplemented with RNAse inhibitors (Promega, Madison, WI, USA) and protease inhibitors free of EDTA (GoldBio, St. Louis, MO, USA). GTP was omitted in the negative control, and 50 mM EDTA was added as the positive control to disassemble the 60S ribosomal subunits. Reaction mixtures of 100 µL were incubated at 30 °C for 10 min and added on top of 150 µL of a 30% sucrose cushion prepared in the same buffer. Samples were centrifuged at 476,000× *g* for 30 min in an Optima tabletop MX ultracentrifuge with a TLA-100 rotor. The top 200 µL (free fraction) was recovered, and the remaining 50 µL was resuspended in 150 µL of the same buffer. Samples were precipitated with 30% trichloroacetic acid, electrophoresed in 10% Tris-glycine SDS-PAGE, transferred to a PVDF membrane, and immunoblotted with antibodies anti-eIF6 (PA531066; Thermo Scientific; dilution 1:3000) and anti-uL3 (OAAB09713; Aviva Systems Biology, San Diego, CA, USA; dilution 1:500) overnight at 4 °C. Subsequently, the membranes were washed and incubated with an anti-rabbit antibody (31466; Thermo Scientific, Waltham, MA, USA; dilution 1:10,000). The chemiluminescent signal was visualised with an ECL chemiluminescence kit (Amersham Bioscience). Signals were quantified using programme Image J version 1.8.0.

### 2.4. Isothermal Titration Calorimetry (ITC)

All calorimetric measurements were carried out using MicroCal^TM^ iTC_200_ System (GE Healthcare, Northampton, MA, USA) in a temperature range of 25–35 °C. Experiments were performed in a buffer consisting of 50 mM Tris-HCl (pH 8.0), 150 mM NaCl, 100 µM DTT, and 5% glycerol. In the titrations with Mg^2+^-bound nucleotides, both the ligand and protein solutions contained 5 mM MgCl_2_. In the case of titrations with Mg^2+^-free nucleotide, both the ligand and protein solutions were supplemented with 2 mM EDTA to sequester any residual trace of metal ions in solution. Protein samples and ligands were prepared in the exact same buffer prior to each titration. Protein concentration in the cell comprised 30–50 µM GTPase, while the ligand concentration in the syringe was 2–3 mM for GTP, MgGTP, GDP, and MgGDP. To ensure measuring the interaction of the complex Efl1 R1086Q•Sdo1 with the nucleotide, an eight-fold concentration excess of Sdo1 was used compared with that of the enzyme. This resulted in >80% of Efl1 being bound to Sdo1 before nucleotide addition, considering the dissociation constant measured in this work. The titration schedule consisted of 15–20 consecutive injections of the ligand with 5 min intervals between injections and a stirring rate of 700 rpm. The dilution heat of the ligand was obtained by adding the ligand to a buffer solution under identical conditions and with the same injection schedule used for the protein sample. All samples were degassed for 10 min prior to the experiment.

The binding isotherms obtained for the isolated GTPase in the presence of Mg(II) were analysed using a ternary model in which the protein can bind to the free nucleotide (G(D)TP) or that bound to one magnesium ion (MgG(D)TP); additionally, GTP may also exist in the form of Mg_2_GTP, which is not recognised by the protein. The derivation of this coupled-equilibria model can be reviewed in [6,31]. The binding constant (K_b_), the enthalpy change (∆H_b_), and the stoichiometry (n) were obtained with the non-linear routines implemented in Origin 5.0 (OriginLab, Co., Northampton, MA, USA) using a one-binding-site model as described in [32,33]. The heterotropic cooperativity for the binding of guanine nucleotides and Sdo1 to Efl1 R1086Q were calculated as previously described in [6,32]. The cooperative effects of Mg(II) on the binding of guanine nucleotides to Efl1 R1086Q alone and bound to Sdo1 were calculated as follows:κ = K_E-MG_/K_E-G_

κ = K_ES-MG_/K_ES-G_
Δh = ΔH_E-MG_ − ΔH_E-G_
Δh = ΔH_ES-MG_ − ΔH_ES-G_
where subscripts E-MG and E-G stand for the association of the metal-bound and metal-free guanine nucleotide with mutant Efl1, respectively. Corresponding expressions applied to the heterotropic cooperativity of the binding of nucleotides to the preformed complex, Efl1•Sdo1 (ES).

The impact of Sdo1 on the binding of Efl1 R1086Q to guanine nucleotides in the presence of magnesium ions is contained in K_ES-MG_ and ΔH_ES-MG_ relative to the K_E-S_ and ΔH_E-S_ values; analogous ratios report the effects in the absence of Mg(II):κ = K_ES-MG_/K_E-S_
κ = K_ES-G_/K_E-G_
Δh = ΔH_ES-MG_ − ΔH_E-S_
Δh = ΔH_ES-G_ − ΔH_E-G_

The corresponding cooperative entropy change was calculated as:TΔs = Δh − Δg = Δh + RTlnκ

The detailed methodology and equations to compute the changes in the accessible surface area (∆ASA) of the GTPase upon binding to nucleotides and the ∆S energetic-based parameters are reported in [6]. Briefly, the water-accessible areas (ASAs) were computed using programme NACCESS, while the changes in accessible surface area (ΔASA) upon the formation of each GTPase–nucleotide complex were estimated considering the difference between the complex and the sum of the free interacting partners. The calculated polar (ΔASAp) and apolar (ΔASAap) area changes were subsequently used to estimate the free molecular partners. Polar (ΔASAp) and apolar (ΔASAap) area changes were used to calculate the heat-capacity change as if the binding occurred as a rigid-body interaction (ΔCp_rb_) according to:ΔCp_rb_ = α∆ASA_ap_ + β∆ASA_p_

where α = 0.45 cal K^−1^ mol^−1^ (Å^2^)^−1^ corresponds to the heat-capacity change of the buried apolar surface area and β = −0.26 cal K^−1^ mol^−1^ (Å^2^)^−1^ represents the heat-capacity change of the buried polar surface [34].

The heat capacity due to conformational changes (ΔCp_conf_) was solved by clearing the variable from:ΔCp_b_ = ΔCp_rb_ + ΔCp_conf_

and the corresponding change in accessible surface area given by the conformational change was solved as follows:ΔCp_conf_ = 0.45ƒ_ap_ ∆ASA_conf_ − 0.26ƒ_ap_ ∆ASA_conf_

where ƒ_ap_ = 0.68 represents the average hydrophobicity index found at protein–protein interfaces [35].

The entropic contribution due to conformational change (ΔS_conf_) was solved by clearing the variable from:ΔS_b_ = ΔS_solv_ + ΔS_r-t_ + ΔS_conf_


Solvation entropy (ΔS_solv_) represents the changes in the degrees of freedom of the solvent molecules upon complex formation and can be calculated as follows:ΔS_solv_ = ΔCp_b_(lnT/Ts)
with Ts being the reference temperature (112 °C) at which the contribution of solvation effects to the overall binding entropy are negligible. The contribution due to changes in roto-translational entropy (ΔS_r-t_) for a bimolecular association corresponds to −8 cal mol^−1^ K^−1^ [36].

## 3. Results

EFL1 works together with its effector protein, SBDS, to release eIF6 in the final maturation step of the 60S ribosomal subunit. Recently, mutations in EFL1 together with those in SBDS have been described to cause Shwachman–Diamond Syndrome [15,17,18]. We enquired if yeast mutation R1086Q, equivalent to the R1095Q disease mutation in human EFL1, could efficiently release Tif6 (the yeast orthologue of human eIF6) using an ex vivo system with purified yeast components (Figure 1). In the presence of wild-type Efl1, Tif6 was enriched in the free fraction, confirming its efficient release by contrasting with the lesser amount that remained bound to the 60S subunits. Oppositely, mutant Efl1 R1086Q resulted in a reduced portion of Tif6 in the free fraction compared with that for the wild type, evidencing its inability to adequately free Tif6. These results, together with previous reports showing similar protein expression levels of the wild type and this mutant in yeast cells, and comparable protein fold and hydrodynamic behaviour [17], reinforce the idea that this mutant is folded but functionally defective. However, the underlying cause of this malfunction is not clear. To shed light on the affected mechanisms, we studied the interaction of this mutant with its target molecules, guanine nucleotides, SBDS, and the 60S ribosomal subunit.

To measure the studied equilibria and not the apparent values due to more than one equilibrium occurring because of catalysis, we measured the interactions between MgGTP, and Efl1 wild type, R1086Q, and a catalytically inactive mutant when present in complex with Sdo1 and the 60S subunit. As a first step, we measured the dependence of GTP hydrolysis mediated by Efl1 R1086Q as a function of Sdo1 and 60S subunits to ensure full occupancy of the GTPase (Appendix A). GTP hydrolysis approached a maximum at a molar ratio of three molecules of Sdo1 per enzyme, suggesting that Efl1 had reached saturation. A second titration of 60S subunits to a constant ratio of 1:3 of GTPase and Sdo1 showed saturation at ~1.5 molecules of enzyme per 60S subunit; thus, a molar ratio of 1:3:1.5 was used to assemble ternary complex GTPase•Sdo1•60S.

We measured the binding of different Efl1 constructs present in complex with Sdo1 and 60S subunits to the fluorescence GTP analogue, mant-GTP, by following the increase in the fluorescence signal after excitation at 355 nm. Fluorescence intensities were plotted as a function of increasing nucleotide concentrations, and the dissociation constant (K_d_) was calculated by fitting the data to a one-binding-site model (Appendix A). The calculated Kd for the binding of mant-GTP to catalytically inactive Efl1 mutant T33A present in complex with Sdo1 and the 60S subunits was 24 μM, a value comparable to those of 36 and 32 μM obtained for the Efl1 wild type and R1086Q mutant, respectively. This indicates that the effect of GDP present in the reaction cell due to catalysis did not skew the calculated dissociation constants in the span of the experimental measurement and supported the subsequent interaction experiments. Furthermore, the presence of mutant Efl1 R1086Q in complex with Sdo1 and 60S subunits did not substantially alter its affinity for GTP in comparison to that bound only to Sdo1 whose dissociation constant at 25 °C corresponded to 29 µM (see below). Because these values were obtained from equilibrium experiments, they implicitly contain the contribution of the conformational dynamics of the complex and thus suggest that Efl1 does not undergo further conformational changes on the ribosome when bound to GTP aside from that previously triggered by Sdo1.

### 3.1. Binding Energetics of Mutant Efl1 R1086Q and Guanine Nucleotides

To characterise the interaction energetics of Efl1 R1086Q with guanine nucleotides and evaluate the contribution of the magnesium ions to the binding, we performed isothermal titration calorimetric studies. Figure 2A,B show representative traces of the binding isotherms obtained for the complexes characterised in this section. Table 1 summarises the thermodynamic binding values for the interaction of Efl1 R1086Q with guanine nucleotides as a function of temperature. It stood out that the thermodynamic processes governing the interactions of mutant R1086Q with both nucleotides were entirely different. The binding to GTP and MgGTP was an endothermic process, while that to GDP was exothermic irrespective of the metal’s presence. This observation contrasts with the thermodynamic signature for the binding of wild-type Efl1 to GTP/MgGTP, which is described by an exothermic process [6]. The binding affinity for both nucleotides was similar even though their thermodynamic signatures differed. The binding to the triphosphate nucleotide was entropically driven, although the binding to the free-metal form exhibited a significant temperature dependence. At 35 °C, the affinity for GTP was three times larger than at 25 °C due to a larger entropic contribution, while in the same temperature interval, that for MgGTP only increased 1.2-fold. In contrast, the temperature had a lesser impact on the binding affinity of the diphosphate nucleotide, with disparate effects on the individual thermodynamic contributions. The interaction with GDP was enthalpically and entropically favourable at 25 °C, and as temperature increased, the entropic contribution became unfavourable and was compensated by an increase in binding enthalpy of ~6 kcal mol^−1^. A similar trend was observed for the binding to MgGDP, with a greater decrease in entropy that remained favourable, compensated by a minor increase in enthalpy of only ~3 kcal mol^−1^.

The mutual effect of nucleotides and Mg(II) on the binding to Efl1 R1086Q is contained in the magnitude of the cooperative heterotropic association constant (*k*), which is defined by the ratio of the metal-bound to the metal-free nucleotide association constants. Magnesium ions destabilised the interaction of Efl1 R1086Q with both nucleotides, as shown for *k* values smaller than 1 (Table 2). This behaviour opposed that observed for wild-type Efl1, in which magnesium ions increased the affinity for GTP (*k* = 2 at 30 °C) [6]. The discrepancy mainly comes from less favourable entropy, suggesting that differential desolvation effects and/or changes in the flexibility of the mutant and wild-type protein impact the interaction affinity. So, they must likely adopt unalike conformations.

We calculated the heat-capacity changes (∆Cp_b_) to investigate the conformational rearrangements of Efl1 R1086Q upon binding to the guanine nucleotides. In protein–ligand interactions and in protein folding, changes in the fraction of polar and nonpolar surfaces buried upon complex formation, together with the degree of surface solvation, relate to the change in the heat capacity of binding and can be easily measured using ITC [37]. The solvation or exposure of polar groups characterises a negative ∆Cp. In contrast, the burial of polar groups results in a positive value and is the signature of the hydrophobic effect driving protein folding. Upon complex formation, the magnitude of Cp may change depending on which phenomenon dominates the process [38,39]. Assuming a linear dependence of ∆H_b_ with the temperature, the analysis of the data resulted in very different heat-capacity changes for the complexes formed by Efl1 R1086Q bound to either nucleotide irrespective of the metal (Figure 2C,D and Table 1). ∆Cp_b_ for Efl1 R1086Q•MgGTP was 10 cal mol^−1^ K^−1^, and it was −270 cal mol^−1^ K^−1^ for the complex with MgGDP. Similar changes were obtained for the corresponding Mg(II)-free nucleotides (30 and −250 cal mol^−1^ K^−1^ for GTP and GDP, respectively). The significantly negative ∆Cp_b_ for the interaction of Efl1 R1086Q with the diphosphate nucleotide indicated the exposure of polar groups to the solvent. In contrast, the conformation adopted by the mutant bound to GTP and MgGTP exposed nonpolar residues to the solvent. Thus, Efl1 R1086Q adopted two distinct conformations depending on the bound nucleotide. More notorious is the difference in ∆Cp_b_ of this mutant compared with wild-type Efl1, whose binding to GTP in the presence and absence of Mg(II) corresponds to a large and negative value [6], reinforcing the idea that mutant Efl1 adopts a different conformation in complex with the triphosphate nucleotide.

∆Cp_b_ can be dissected into two contributions, one concerning the conformational changes undergone by interacting molecules (∆Cp_conf_) and the other related to the amount of hidden area at the binding interface as if the binding occurred as a rigid-body association (∆Cp_rb_) [40]. Similarly, the entropy of binding (∆S_b_) consists of the contribution of three components: solvation entropy (∆S_solv_), roto-translational entropy (∆S_r-t_), and conformational entropy (∆S_conf_) [36,41]. The first refers to changes in the solvation of the interacting surfaces, while the second results from the restriction of the rotation and translation motions of the binding molecules. The deconvolution of measured ∆Cp_b_ and ∆S_b_ into their corresponding conformational contributions helps to rationalise the significance of these thermodynamic parameters in molecular terms. ∆Cp_b_ for GDP/MgGDP was mainly dominated by the contribution of the conformational change in which Efl1 R1086Q buried a surface of ~1000 Å corresponding to 21 residues (Table 3). In contrast, binding to GTP/MgGTP reflected the counterbalance of opposing contributions from the rigid-body association and the conformational heat-capacity change. The first one involved the burial of approximately three nonpolar residues at the binding interface and the exposure of three other nonpolar residues arising from the conformational change. Comparing these data with previous reports for the wild-type protein suggests that both proteins undergo a similar conformational change upon binding to the diphosphate nucleotide in the absence of Mg(II). However, seemingly dissimilar for the triphosphate nucleotide, considering the twenty-fold excess of residues (66 vs. 3 residues) being buried in the native enzyme. The extent of these conformational changes in Efl1 R1086Q were similar when analysed in terms of conformational entropy. Many routable bonds stiffened in the complex with GDP/MgGDP, with just as many loosening in the complex bound to GTP/MgGTP. Favourable solvation entropy for the complex of Efl1 R1086Q with GDP/MgGDP contrasting with the slightly unfavourable value for GTP/MgGTP suggested a different rearrangement of the solvent molecules around the complexes due to the exposure of hydrophobic groups in the latter (Table 3). The conformation adopted by Efl1 R1086Q was different when bound to either nucleotide, but it paralleled that of the wild type in complex with GDP/MgGDP. The mutation profoundly affected the recognition of GTP/MgGTP with a larger affinity at the expense of an altered conformational change.

Early studies on C-terminal deletions of Efl1 identified intragenic suppressor mutations that correct the slow-growth phenotype conferred by the removal of the last 22 residues of the protein [42]. We reasoned that any of these two mutations, P151L or T657M, may alleviate the defects described above for the binding of Efl1 R1086Q to MgGTP. Double mutant Efl1 R1086Q P151L exhibited a partial recovery of the thermodynamic parameters of the binding to MgGTP with respect to the Efl1 wild type (Table 1 and Table 3). More notorious is the regaining of an exothermic process driven by both contributions, enthalpy and entropy, at all the temperatures tested, as previously observed for wild-type Efl1 [6]. ∆Cp_b_ of −220 cal mol^−1^ K^−1^ for the double mutant contrasted with the positive value obtained for R1086Q but approximated the −560 cal mol^−1^ K^−1^ of the wild-type enzyme. The deconvolution of the heat capacity resulted in half of the polar residues being exposed to the solvent compared with those in wild-type Efl1. Thus, suppressor mutation P151L present in domain I seemed to partially rescue the altered conformation elicited by mutation R1086Q located far away in domain IV of the yeast Efl1 orthologue.

### 3.2. Effect of Sdo1 in the Interaction Energetics of Mutant Efl1 R1086Q with Guanine Nucleotides

It has been proposed that Sdo1 increases the affinity of Efl1 for GTP by inducing a conformational change that favours the binding of this nucleotide [5,6]. We investigated if Sdo1 could elicit the same effect on the Ef11 R1086Q mutant by measuring their interaction and the consequence of this interaction on guanine-nucleotide affinity (Figure 3 and Table 4). The affinity of Sdo1 for Efl1 R1086Q increased two-fold with the temperature and was enthalpically driven, with favourable entropy at 25 and 30 °C that became unfavourable at 35 °C. However, these binding parameters largely differed from those reported for the wild-type complex. The affinity of Efl1 R1086Q for Sdo1 decreased by one order of magnitude from 0.3 µM to 3 µM with respect to that of native Efl1 (Table 4). The analysis of the thermal dependence of ΔH_b_ resulted in a ΔCp_b_ of −440 cal mol^−1^ K^−1^ for the complex with mutant Efl1 (Table 3), which corresponded to half the magnitude of that for wild-type Efl1 (−860 cal mol^−1^ K^−1^). This difference suggests that the two complexes differ in half the number of residues being buried at the protein–protein interface or a change in conformation.

In subsequent experiments, the pre-formed Efl1 R1086Q•Sdo1 complex was titrated with guanine nucleotides (Table 4). The binding to the triphosphate was both enthalpically and entropically driven, with the magnesium ions having a large impact on the thermal dependence of these parameters. In the presence of magnesium ions, binding entropy decreased as a function of the temperature compensated by more favourable enthalpy, while in their absence, both contributions remained almost unchanged. This disparate effect of the metal on the binding of the complex for the triphosphate nucleotide resulted in dissimilar heat-capacity changes, −400 cal mol^−1^ K^−1^ for MgGTP, contrasting with 30 cal mol^−1^ K^−1^ for GTP, which in turn implied the exposure of a different number of residues (36 vs. 4) of opposing hydrophobic nature (Table 3) and the consequent difference in conformation. Similarly, the dissection of binding entropy depicted a favourable solvation component but an unfavourable conformational change upon interaction with MgGTP, indicating a large change in the degrees of freedom of the solvent molecules released during complex formation. These contributions were reverted in the absence of magnesium ions, reinforcing the idea that the metal impacted the recognition of this nucleotide and the corresponding conformational change (Table 3). The inspection of the cooperativity parameters showed that magnesium ions and Sdo1 destabilised the interaction of Efl1 R1086Q with the triphosphate nucleotide (κ ≤ 1) due to unfavourable entropy (Table 2). This observation largely differed from the effect Sdo1 had on wild-type Efl1, which favoured 20–40-fold the interaction with GTP [6]. More notorious is the drastic change in binding enthalpy for the triphosphate from an endothermic process for the isolated enzyme to an exothermic one for the Efl1 R1086Q•Sdo1 complex. In the absence of magnesium ions, the affinity of the preformed complex for GTP decreased by 2–4 times compared with the free enzyme due to a large discrepancy between the less favourable entropy compensated by the enthalpic contribution (Table 1 and Table 4). The heat-capacity change of 30 kcal mol^−1^ K^−1^ implied that only ~four residues changed conformation between the free and Sdo1-bound enzymes. In the presence of the metal, the affinity of complex Efl1 R1086Q•Sdo1 for MgGTP decreased as the temperature raised and showed a favourable enthalpic component with respect to that of the isolated enzyme. These differences in the thermodynamic contributions resulted in very different heat capacities for Sdo1-bound Efl1 R1086Q and the free enzyme (−400 cal mol^−1^ K^−1^ vs. 10 cal mol^−1^ K^−1^) with the concomitant differences in the number and nature of residues changing conformation in both complexes (Table 3). These results suggested that the complex formed between Efl1 R1086Q•Sdo1 and the triphosphate nucleotide and their thermodynamic properties were influenced by magnesium ions and were different from those of the wild-type enzyme. Similarly, at 25–30 °C, the binding of the diphosphate nucleotide to the Efl1 R1086Q•Sdo1 complex was also driven by favourable enthalpic and entropic contributions. As previously discussed, magnesium ions also elicited disparate effects on the binding of this nucleotide, and the affinity was 2–5 times greater in the presence of magnesium ions due to a more favourable entropic contribution (Table 4). The dissection of binding entropy for both GDP and MgGDP indicated a negative contribution from conformational entropy compensated by the solvation entropy, albeit to different extents. In the absence of magnesium, twice more residues were buried with an equivalent number of routable bonds freezing in the preformed complex (Table 3). The analysis of the mutual cooperativity indicated that the favourable entropic effect of Sdo1 on the interaction of mutant Efl1 with the diphosphate nucleotide depended on the presence of magnesium ions (Table 2).

Reciprocally, the diphosphate decreased three-fold the interaction of Sdo1 with mutant Efl1 in the presence of magnesium ions. In contrast, the triphosphate had no impact on the interaction between the two proteins (Figure 4). Therefore, Sdo1 aggravated the interaction of mutant Efl1 with MgGTP by an order of magnitude, a dissenting effect expected for Sdo1 acting as an effector that drives Efl1 into the active conformation, as it was previously described for the wild-type protein. The interaction of wild-type Efl1 with Sdo1 increased the affinity of the GTPase for its substrate GTP by a factor of ~70, predisposing a conformation in Efl1 that made it readily accessible to accommodate GTP with no further changes in ΔASA in relation to the isolated protein. In contrast, the association of MgGTP with preformed mutant Efl1 R1086Q•Sdo1 was accompanied by large and dissimilar ΔCp_b_ and ΔH_b_ compared with those observed for the isolated enzyme (Table 1 and Table 4). This implied that a large conformational change still happened upon binding (ΔΔCp_b_ −410 cal mol^−1^ K^−1^) with the additional modification of 40 residues with respect to the isolated enzyme (Figure 2). On the other hand, the associated ΔCp_b_ value of MgGDP binding to mutant Efl1 was similar in the presence and absence of Sdo1 (ΔΔCp_b_ −45 cal mol^−1^ K^−1^) suggesting the burial of only four additional residues (Figure 2). Interestingly, the effect Sdo1 elicited on the conformational changes of mutant Efl1 upon binding to guanine nucleotides depended on the presence of magnesium ions, implying that Sdo1 may exert its influence by remodelling the metal binding pocket. Together the results showed that mutation R1086Q profoundly affected the internal rearrangement of interactions necessary to elicit the same conformational change observed in native Efl1 in response to its effector ligands.

Finally, we measured the enzyme kinetics of mutant Efl1 R1086Q as an isolated enzyme and bound to Sdo1 using an enzymatic coupled assay with a purine nucleoside phosphorylase (PNP) enzyme [43]. This analysis was extended to another disease mutation (EFL1 M882K) also located in domain IV of the GTPase using the equivalent yeast variant, Efl1 L910K [17]. The *K_m_* and *k*_cat_ kinetic parameters were calculated using non-linear regression assuming a Michaelis–Menten model (Appendix A). Both kinetic variables were almost identical for the R1086Q mutant with respect to the wild-type enzyme, while those for the L910K mutant showed a modest decrease of 13% in *k_cat_* and a similar increase in *K_m_*. The catalytic constant for mutant R1086Q and the wild-type enzyme consisted of 0.67 min^−1^, and it was 0.58 min^−1^ for mutant Efl1 L910K. The corresponding values of the catalytic constants for the binary complexes of mutant GTPases and Sdo1 did not significantly change with respect to the wild-type protein complex. Together, these results confirmed that these mutations did not alter the catalytic properties of Efl1. In line with the proposed role of Sdo1 as a guanine exchange factor of Efl1, previous results have shown that the interaction of wild-type Efl1 with Sdo1 impacted its *K_m_* for GTP, decreasing its value 2–3-fold [5,6,44]. The *K_m_* values for the complexes formed between Sdo1 and Efl1 mutants decreased by half with respect to their cognate-free counterpart and differed from each other. The Sdo1 complexes with Efl1 mutants R1086Q and L910K had *K_m_* values approximately 1.6 larger than that of the native protein, consisting of 84 and 90 µM, respectively, and contrasting with the value of 53 µM for wild-type Efl1. The measured *K_m_* value comprises all the microscopic rate constants of the individual reactions occurring in the enzyme, which include the binding to the substrate and product and the corresponding conformational changes. Thus, the increase in *K_m_* for the mutant binomials suggested that their recognition of GTP was impaired largely because Sdo1 could not elicit the conformational change occurring in the native complex.

## 4. Discussion

Conformational changes are central to the function and regulation of GTPases, which transit between a compact, active conformation bound to GTP and a more elongated, inactive structure in complex with GDP [45]. The structural rearrangements inferred from the heat capacity and entropy of mutant Efl1 did not resemble the trend observed for translational GTPases or that of wild-type Efl1. The thermodynamic signature of the interaction between mutant Efl1 and MgGDP/GDP was different from that described for the wild-type protein, implying that the complexes are not alike. No conformational changes occurred upon binding to MgGTP, as judged by the small heat-capacity change close to zero and the burial of only 3 residues, a small number compared with the 51-residue changing conformation in the wild-type complex. So, mutant Efl1, instead of adopting an active T-conformation upon binding to MgGTP, remained in the initial form. This *apo* conformation may not necessarily be the same between mutant R1086Q and the wild-type protein, since differences in their radius of gyration (*R_g_*) have already been reported [17]. The heterotropic cooperative parameters indicated that Mg(II) weakened the interaction of mutant Efl1 R1086Q with GTP and strengthened it for GDP, an opposing trend with respect to that described for classical GTPases.

GTPases operate as molecular switches with sophisticated regulatory mechanisms that sense the bound nucleotide through the interaction with effector biomolecules such as guanine nucleotide exchange factors (GEFs) and GTPase-activating proteins (GAPs) [46]. The proposed role of Sdo1/SBDS as a GEF of Efl1/EFL1 is to facilitate the transition of the GTPase into the active T-conformation favouring the interaction with GTP [6]. Instead, the interaction of Sdo1 with mutant Efl1 did not trigger any conformational change in the GTPase. ΔCp of −440 cal mol^−1^ K^−1^ for the two vertical branches of the GTP thermodynamic cycle (Appendix A) in the absence of Mg(II) ions corresponded to the burial of the protein–protein interface and/or the conformational change in Sdo1 but had no contributions from a conformational change happening in the GTPase. Thus, the additional heat capacity observed for the interaction of complex Efl1 R1086Q•Sdo1 with MgGTP (ΔCp −410 cal mol^−1^ K^−1^) was contributed by a conformational change occurring exclusively in the GTPase. Whether this shape resembles the T-conformation achieved by the wild-type protein is still unclear, as these data were obtained only in the absence of the metal, and it has already been noted that Mg(II) has a profound impact on the interaction. However, the overall heat-capacity change of −950 cal mol^−1^ K^−1^ required to drive the native protein into ternary complex GTPase•GTP•Sdo1 was twice larger than that in mutant Efl1 (−410 cal mol^−1^ K^−1^). These differences suggest that the conformations of the mutant and wild-type proteins bound to Sdo1 and the triphosphate are different, and probable so is their function. Binding to MgGDP triggered a similar conformational change in isolated mutant Efl1 and that in the pre-formed complex with Sdo1. In the absence of magnesium ions, the free mutant and that bound to Sdo1 adopted two discernible conformations differing by twice the number of residues, an analogous scenario previously described for the wild-type protein and GDP. Based on these observations, it is evident that Sdo1 communicates with structural elements of switches 1 or 2 to mediate the conformational changes and modify the affinity of Efl1 for guanine nucleotides. Several GEFs remodel these switches through hydrophobic repulsion in the Mg(II) binding pocket [46]. The data shown here could imply that it is the case for this GTPase. However, in the current structures (PDB 5ANB, 5ANC), SBDS locates far away from the G-subdomains, which may only be explained by a long-distance allosteric effect. A summary of the conformational changes occurring in Efl1 R1086Q in response to its interaction with Sdo1 and nucleotides are shown in Figure 4 and Appendix A. The present analysis suggests that mutant Efl1 R1086Q in solution adopts three differentiable conformations: 1) an *apo* form in the absence of any ligand and bound to MgGTP or Sdo1 (Efl1 R1086Q_apo_), 2) a *D-like* conformation that can be achieved via the interaction with MgGDP or Sdo1 (Efl1 R1086Q_D_), and 3) a *T*-like* conformation that differs from that of the wild-type protein (Efl1 R1086Q_T*_). Therefore, this mutant exhibited a complex conformational landscape mediated by the interaction with both nucleotides and Sdo1, as previously described for the native protein, albeit differently. Its interaction with guanine nucleotides and Mg(II) did not resemble that of the wild type, and Sdo1 no longer induced the energetic and structural effects that favoured the interaction of the mutant GTPase with the trinucleotide. On the contrary, it increased the relative affinity of the mutant GTPase for GDP by a factor of twelve, driving the protein to the inactive conformation. Taken together, the results indicate that mutation R1086Q in domain IV perturbs the long-distance intramolecular communication that senses the nucleotide-binding status occurring in domain I of Efl1. These alterations in the thermodynamic signature of mutant Efl1 could explain its malfunction and aid our understanding of the molecular causes of SDS.

To what extent this altered conformational plasticity is transferable and detrimental in the ribosomal context awaits further investigation. However, it is difficult to envisage that the deviations observed here are fortuitous, since this mutant cannot efficiently release Tif6 (Figure 1), and the conformational transition facilitated by Sdo1 is vital for the eviction. Upon binding to the ribosomal subunit, Efl1 mostly adopt a T-like conformation stabilised by Sdo1, triggering the release of Tif6. The subsequent hydrolysis of GTP drives Efl1 into a GDP-bound conformation, prompting the release of the GTPase and its cofactor, Sdo1 [7]. According to our calorimetric analyses, only a small proportion of mutant Efl1 can be recruited to the ribosomal subunit by Sdo1, as expected by the one-order-of-magnitude decrease in the affinity of the two proteins. Instead of stabilising the GTP-bound conformation, Sdo1 favours the GDP-bound form of Efl1 R1086Q, a non-productive conformation unable to release Tif6 and affecting the rate of 60S subunits incorporated into the pool of translating ribosomes. If the upregulation of Efl1 is attenuated, the cells no longer have a persistent signal to produce mature ribosomal subunits that fulfils the cell requirements for translation.

## Figures and Tables

**Figure 1 biomolecules-12-01141-f001:**
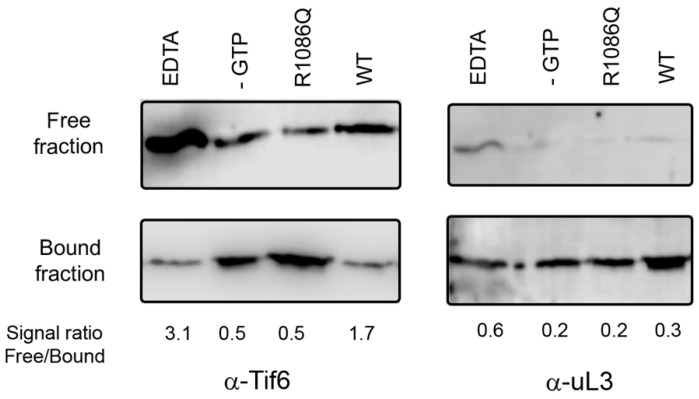
Tif6 release assay mediated by yeast Efl1. Yeast 60S ribosomal subunits were incubated in the presence of GTP, Sdo1, and yeast Efl1 wild-type or R1086Q mutant and sedimented through a 30% sucrose cushion. The distribution of Tif6 in the supernatant (free fraction) or sedimented together with the 60S subunit (bound fraction) was visualised using immunoblot. Positive control contained EDTA to disassemble the ribosomal subunits. Negative control lacked GTP.

**Figure 2 biomolecules-12-01141-f002:**
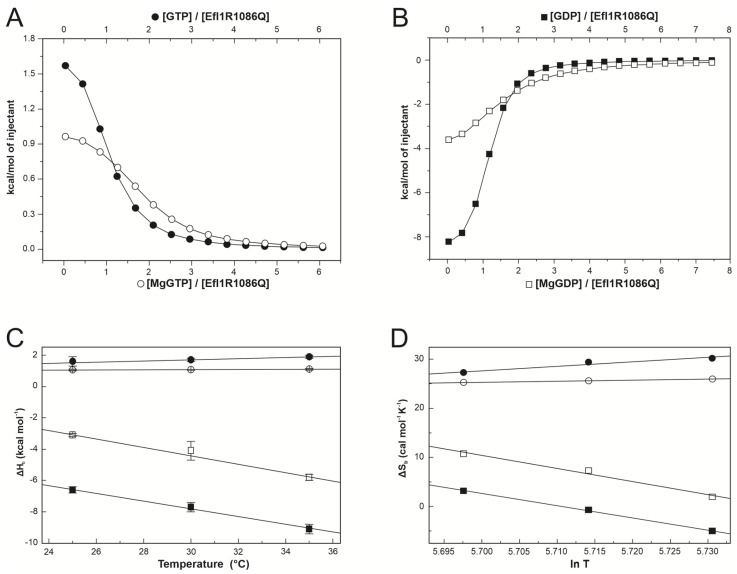
Calorimetric characterisation of the interaction between Efl1 R1086Q and guanine nucleotides. Panels (**A**,**B**) depict representative binding isotherms obtained at 30 °C for GTP (●), MgGTP (○), GDP (■), and MgGDP (□). Solid lines correspond to the fit to a one-binding-site model using non-linear regression. Panels (**C**,**D**) present binding enthalpy and entropy as functions of temperature. Solid lines correspond to the least squares linear fitting describing the thermal dependence of ∆H_b_ or ∆S_b_ assuming ∆Cp_b_ is constant in the temperature range spanned.

**Figure 3 biomolecules-12-01141-f003:**
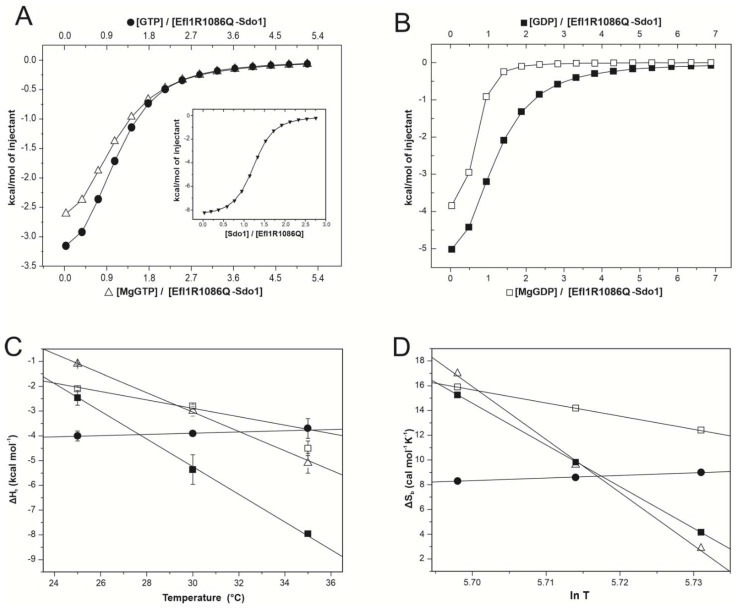
Calorimetric characterisation of the interaction between complex Efl1 R1086Q•Sdo1 and guanine nucleotides. Panels (**A**,**B**) depict representative binding isotherms obtained at 30 °C for GTP (●), MgGTP (△), GDP (■), and MgGDP (□). Insertion in panel A corresponds to the binding of Efl1 R1086Q to Sdo1. Solid lines correspond to the fit to the one-binding-site model using non-linear regression. Panels (**C**,**D**) present binding enthalpy and entropy as functions of temperature. Solid lines correspond to the least squares linear fitting describing the thermal dependence of ∆H_b_ or ∆S_b_ assuming ∆Cp_b_ is constant in the temperature range spanned.

**Figure 4 biomolecules-12-01141-f004:**
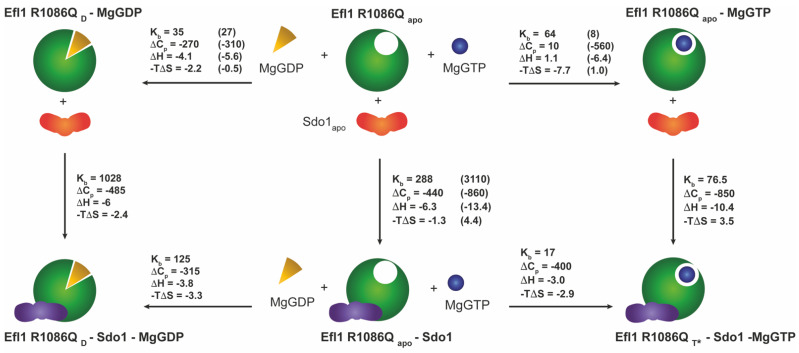
Energetics of the coupled equilibria among mutant Efl1 R1086Q, Sdo1, and guanine nucleotides in the presence of magnesium ions at 30 °C. Analyses indicated that mutant Efl1 adopted three different conformations: (1) an *apo* form of the enzyme adopted by the free enzyme and bound to MgGTP or Sdo1 (Efl1 R1086Q_apo_), (2) a *D* conformation (Efl1 R1086Q_D_) bound to MgGDP or MgGDP•Sdo1, and (3) a *T** conformation interacting with MgGTP and Sdo1. Data in parenthesis correspond to those of wild-type Efl1 as reported in [6]. Units: K_b_—mM^−1^; -TΔS and ΔH—kcal mol^−1^; ΔCp_b_—cal mol^−1^ K^−1^.

**Table 1 biomolecules-12-01141-t001:** Thermodynamic parameters for the binding of Efl1 R1086Q to guanine nucleotides determined using isothermal titration calorimetry.

Ligand		T (°C)	K_b_ (mM^−1^)	K_d_ (μM)	ΔG_b_ (kcal mol^−1^)	ΔH_b_ (kcal mol^−1^)	-TΔS_b_ (kcal mol^−1^)	ΔCp_b_ (cal mol^−1^ K^−1^)
Efl1 R1086Q								
GTP		25	61 ± 7	16 ± 2	−6.53	1.04 ± 0.05	−7.57	
	+Mg^2+^	30	64 ± 11 (8 ± 0.7)	16 ± 3 (125 ± 11)	−6.67 (−5.4)	1.07 ± 0.03 (−6.4 ± 0.4)	−7.74 (1.0)	10 ± 2 (−560 ± 23)
		35	76 ± 10	13 ± 2	−6.88	1.12 ± 0.06	−8.0	
		25	60 ± 18	17 ± 5	−6.52	1.6 ± 0.3	−8.12	
	−Mg^2+^	30	156 ± 34 (4 ± 0.3)	6 ± 1 (250 ± 19)	−7.20 (−5.0)	1.7 ± 0.1 (−7.3 ± 0.3)	−8.90 (2.3)	30 ± 6 (−700 ± 34)
		35	178 ± 28	6 ± 1	−7.40	1.9 ± 0.1	−9.30	
GDP		25	41 ± 5	24 ± 3	−6.29	−3.1 ± 0.1	−3.19	
	+Mg^2+^	30	35 ± 7 (27 ± 3)	28 ± 6 (37 ± 4.1)	−6.30 (−6.1)	−4.1 ± 0.6 (−5.6 ± 0.2)	−2.20 (−0.5)	−270 ± 40 (−310 ± 5)
		35	34 ± 9	29 ± 8	−6.39	−5.8 ± 0.2	−0.59	
		25	336 ± 47	3 ± 0.4	−7.54	−6.6 ± 0.2	−0.94	
	−Mg^2+^	30	245 ± 37 (64 ± 9)	4 ± 0.6 (16 ± 2.2)	−7.47 (−6.7)	−7.7 ± 0.3 (−6.7 ± 0.3)	0.22 (0.0)	−250 ± 17 (−318 ± 35)
		35	226 ± 34	4 ± 0.6	−7.55	−9.1 ± 0.3	1.55	
Efl1 R1086Q P151L								
GTP		25	55 ± 10	18 ± 3	−6.5	−1.7 ± 0.1	−4.8	
	+Mg^2+^	30	59 ± 8	17 ± 2	−6.6	−2.5 ± 0.1	−4.1	−220 ± 35
		35	54 ± 5	18 ± 2	−6.7	−3.9 ± 0.1	−2.8	

Data in parenthesis correspond to those of wild-type Efl1 as reported in [6].

**Table 2 biomolecules-12-01141-t002:** Effect of Mg^2+^ and Sdo1 in the cooperativity parameters for the heterotropic interaction of Efl1 R1086Q with guanine nucleotides.

Complex	Ligand	T (°C)	κ	Δg (kcal mol^−1^)	Δh (kcal mol^−1^)	-TΔs (kcal mol^−1^)
Effect of Mg^2+^						
Efl1 R1086Q + Sdo1	GTP	25	0.6	0.3	2.9	−2.6
		30	0.3	0.6	0.9	−0.3
		35	0.5	0.5	−1.4	1.9
	GDP	25	1.9	−0.4	0.3	−0.7
		30	3.7	−0.8	1.6	−2.4
		35	5.4	−1.0	2.7	−3.7
Efl1 R1086Q − Sdo1	GTP	25	1.0	−0.1	−0.56	0.5
		30	0.4 (2)	0.6 (−0.4)	−0.63 (0.9)	1.2 (−1.3)
		35	0.4	0.5	−0.78	1.3
	GDP	25	0.1	1.3	3.5	−2.3
		30	0.1 (0.4)	1.2 (0.6)	3.6 (1.1)	−2.4 (−0.5)
		35	0.2	1.2	3.3	−2.1
Effect of Sdo1						
Efl1 R1086Q + Mg^2+^	GTP	25	0.6	0.3	−2.1	2.5
		30	0.3	0.7	−4.1	4.8
		35	0.3	0.9	−6.2	7.1
	GDP	25	2.7	−0.6	1.0	−1.6
		30	3.6	−0.8	0.3	−1.1
		35	4.6	−0.9	0.5	−1.4
Efl1 R1086Q − Mg^2+^	GTP	25	1.0	0.0	−5.6	5.6
		30	0.3 (22.3)	0.7 (−1.8)	−5.6 (5.2)	6.3 (−7.0)
		35	0.2	0.9	−5.6	6.5
	GDP	25	0.2	1.06	4.2	−3.1
		30	0.1 (0.3)	1.2 (0.8)	2.3 (−0.7)	−1.2 (1.5)
		35	0.1	1.3	1.2	0.5

Data in parenthesis correspond to or were calculated from those reported in [6] for wild-type Efl1.

**Table 3 biomolecules-12-01141-t003:** Energy-based calculations of the conformational changes of mutant Efl1 alone or in complex with Sdo1 upon nucleotide binding at 30 °C.

Ligand		ΔCp_b_	ΔCp_rb_	$\Delta Cp_{conf}^{}$	$-\Delta ASA_{conf}^{}$	${N_{conf}^{res}}_{\;}$	ΔS_b_	ΔS_solv_	ΔS_conf_	$N_{conf}^{rotb}$
Efl1 R1086Q										
GTP	+Mg^2+^	10 (−560)	−25 (−25)	35 (−535)	−157 (2401)	3 (51)	25 (−3)	−2 (143)	36 (−139)	21 (83)
	−Mg^2+^	30 (−700)	−11 (−11)	41 (−689)	−184 (3092)	4 (66)	29 (−8)	−7 (179)	44 (−179)	27 (107)
GDP	+Mg^2+^	−270 (−310)	−48 (−48)	−222 (−262)	996 (1176)	21 (25)	7 (2)	65 (79)	−49 (−70)	30 (42)
	−Mg^2+^	−250 (−318)	−30 (−30)	−220 (−288)	987 (1293)	21 (28)	−1 (0)	60 (81)	−53 (−73)	31 44)
Efl1 R1086Q•Sdo1										
GTP	+Mg^2+^	−400	−25	−375	1683	36	9	96	−78	47
	−Mg^2+^	30 (−90)	−11 (−11)	41 (−79)	−184 (355)	4 (8)	9 (16)	−7 (23)	24 (0)	14 (0)
GDP	+Mg^2+^	−315	−48	−267	1198	25	11	75	−57	34
	−Mg^2+^	−550 (−640)	−30 (−30)	−520 (−610)	2334 (2738)	50 (58)	3 (−5)	132 (164)	−121 (−161)	72 (96)
Efl1 R1086Q P151L										
GTP	+Mg^2+^	−220	−25	−195	875	19	13	53	−31	19

Units for ∆Cp and ∆S correspond to cal mol^−1^ K^−1^ and for ΔASA are Å^2^. $N_{conf}^{res}$ corresponds to the number of residues changing conformation as absolute value. $N_{conf}^{rotb}$ corresponds to the number of routable bonds changing conformation as absolute value. Data in parenthesis correspond to or were calculated from those reported in [6] for wild-type Efl1.

**Table 4 biomolecules-12-01141-t004:** Thermodynamic parameters for the binary and ternary interaction systems formed among Efl1 R1086Q, Sdo1, and guanine nucleotides determined using isothermal titration calorimetry.

Ligand		T (°C)	K_b_ (mM^−1^)	K_d_ (μM)	ΔG_b_ (kcal mol^−1^)	ΔH_b_ (kcal mol^−1^)	-TΔS_b_ (kcal mol^−1^)	ΔCp_b_ (cal mol^−1^ K^−1^)
Efl1 R1086Q								
Sdo1		25	256 ± 47	3.9 ± 0.7	−7.4	−4.3 ± 0.1	−3.1	−440 ± 23 (−860 ± 31)
		30	288 ± 56 (3110 ± 630)	3.4 ± 0.6 (0.3 ±0.0)	−7.6 (−9.0)	−6.3 ± 0.2 (−13.4 ± 0.4)	−1.3 (4.4)	
		35	543 ± 76	1.8 ± 0.2	−8.1	−8.7 ± 0.1	0.6	
Efl1 R1086Q•Sdo1								
GTP	+Mg^2+^	25	34 ± 3	29 ± 5	−6.2	−1.1 ± 0.04	−5.1	−400 ± 11
		30	17 ± 1	58 ± 7	−5.9	−3.0 ± 0.2	−2.9	
		35	19 ± 2	52 ± 7	−6.0	−5.1 ± 0.4	−0.9	
		25	57 ± 6	17 ± 5	−6.5	−4.0 ± 0.2	−2.5	30 ± 6 (−90 ± 5)
	−Mg^2^	30	44 ± 1 (89 ± 12)	23 ± 7 (11 ± 1)	−6.5 (−6.8)	−3.9 ± 0.1 (−2.1 ± 0.2)	−2.6 (−4.7)	
		35	39 ± 8	25 ± 7	−6.5	−3.7 ± 0.4	−2.8	
GDP	+Mg^2+^	25	110 ± 23	9 ± 2	−6.9	−2.1 ± 0.08	−4.7	−315 ± 9
		30	125 ± 14	8 ± 1	−7.1	−3.8 ± 0.05	−3.3	
		35	158 ± 18	6 ± 1	−7.3	−5.3 ± 0.11	−2.0	
	−Mg^2+^	25	57 ± 1.5	17 ± 0.5	−6.5	−2.5 ± 0.4	−4.0	−550 ± 18 (−640 ± 34)
		30	33 ± 5.0 (19 ± 1)	30 ± 4 (52 ± 1)	−6.3 (−5.9)	−5.4 ± 0.6 (−7.4 ± 0.0)	−1.0 (1.5)	
		35	29 ± 4.2	34 ± 5	−6.30	−8.0 ± 0.8	1.7	

Data in parenthesis correspond to those of wild-type Efl1 as reported in [6].

## Data Availability

All data generated in this study are contained within the article or the Appendix A.

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
