# Peer review of "Altered Conformational Landscape upon Sensing Guanine Nucleotides in a Disease Mutant of Elongation Factor-like 1 (EFL1) GTPase"

_biomolecules, 2022, doi:10.3390/biom12081141_

Round 1

Reviewer 1 Report

Ribosomes are assembled through the stepwise addition of ribosomal components. In the final step of the formation of the eukaryotic 60S ribosomal subunit initiation factor IF6 is released from the inactive precursor 60S with the help of EF1-like (Efl1) and its the GDP-GTP exchange factor Sdo1. Mutations in Efl1 and Sdo1 lead to a congenital disease called Schwachman-Diamond syndrome in humans. Some of these mutations reduce the efficiency of the eIF6 release, which affects the cell’s protein synthesis capacity. Other mutations do not impair ribosome production, yet still, cause the syndrome. To shed light on the mechanisms of the latter class of mutants, Sánchez-Puig and coworkers have measured the thermodynamic parameters of an Efl1 mutant protein (Efl1 R1095Q) binding to guanine nucleotides, Sdo1, and their interplay in solution. Based on the difference between the mutant and wildtype thermodynamic parameters, they propose that structural changes mediated by a long-distance network in the wildtype fail to occur in the mutant Efl1 protein. This is an interesting and potentially important suggestion, but the work lacks controls and is not yet suitable for publication.

Specific suggestions

·      The main problem with the paper is that control experiments with the wildtype Efl1 are only shown in the western experiment in Figure 1. Efl1 wildtype controls for the thermodynamic and kinetic experiments are missing entirely from this manuscript. Instead, the authors refer to a previous publication in which Dr. Sánchez-Puig is a co-author (ref 6). Granted, the overlap in authorship assures comparability of the previously published data with the data in this manuscript.  Nevertheless, I strongly suggest that the original, or at least the essential data from the previous paper should be included in the current manuscript (with proper permissions!) so that the reader can directly and easily evaluate the mutant and wildtype without recovering the previous paper. Similarly, the model figure (Figure 4) should include the wildtype values. All wildtype data could be overlaid on the mutant results without expanding the space occupied in the journal.

·      It would be helpful, if the proposed model could be illustrated in the context of the cryo-EM model, possibly with the help of the recently developed AI structure prediction software.

·      Figure 1

o   Please quantify the western blots

o   Why is uL3 a double band in the right-hand panel

·      Line 149-50: “methodologies section of the main article.” Please explain.

Author Response

Manuscript Biomolecules_1847364

Altered conformational landscape upon sensing guanine nucleotides in a disease mutant of the Elongation Factor-like 1 (EFL1) GTPase

Response to reviewer 1

We have reviewed the use of language and have done some amendments to the style and grammar.

Query 1

The main problem with the paper is that control experiments with the wildtype Efl1 are only shown in the western experiment in Figure 1. Efl1 wildtype controls for the thermodynamic and kinetic experiments are missing entirely from this manuscript. Instead, the authors refer to a previous publication in which Dr. Sánchez-Puig is a co-author (ref 6). Granted, the overlap in authorship assures comparability of the previously published data with the data in this manuscript. Nevertheless, I strongly suggest that the original, or at least the essential data from the previous paper should be included in the current manuscript (with proper permissions!) so that the reader can directly and easily evaluate the mutant and wildtype without recovering the previous paper.

Response

Contrasting the data presented here for the mutant Efl1 R0186Q with those previously reported for the wild-type protein in Luviano et. al. is vital to understand the alterations arising by this mutation. As such we agree with the reviewer that presenting some of the data concerning the wild-type within this manuscript would facilitated the reader rather that going back to the previous paper. In this regard, we have added data corresponding to wild-type Efl1 in the corresponding tables along the manuscript. Only the information at 30 °C was added not to clog the tables with more data, as already there is a large amount of information to process for the reader, and the information at 30 °C is representative of the differences between mutant and wild-type proteins and corresponds to the optimal growth temperature of yeast considering we used the yeast Efl1 orthologue.

Wild-type data were added in parenthesis next to the corresponding information of the mutant and explained at the bottom of the corresponding table. In Luviano et. al the interaction of the complex Efl1 WT·Sdo1 with the guanine nucleotides was not evaluated in the presence of magnesium ions, so in Tables 2, 3 and 4 these data are not presented.

Query 2

Similarly, the model figure (Figure 4) should include the wildtype values. All wildtype data could be overlaid on the mutant results without expanding the space occupied in the journal.

Response

We have added the data corresponding to the wild-type Efl1 next to the mutant´s not only in Figure 4 but also in the supplementary figure 2. As mentioned above, only available data was added considering there is no information for the complex Efl1 WT·Sdo1 with the guanine nucleotides in the presence of magnesium ions.

Query 3

It would be helpful, if the proposed model could be illustrated in the context of the cryo-EM model, possibly with the help of the recently developed AI structure prediction software.

Response

Although having a possible model for the conformational changes occurring in the R0186Q mutant would be very helpful, the ITC data does not have such structural resolution. Many combinations of residues being buried and exposed can result in similar DCpb so trying to infer a conformation out of this parameter and its analysis may be misleading and result in the overinterpretation of the data. As we mentioned in the final paragraph of the manuscript, we believe that the conformational alterations elicited by this mutant in solution alone and bound to Sdo1 are also transferred into the ribosomal subunit context but does not mean they have to be identical. Furthermore, the local resolution of Efl1 in the current cryo-EM structures (5ANB and 5NC, Weis et. al.) is very low around 8 Å to further propose an model.

Query 4

Figure 1:

  • Please quantify the western blots
  • Why is uL3 a double band in the right-hand panel

Response

We quantified the signal of each band and presented the number at the bottom of each corresponding sample in Figure 1 as the signal ratio of the free fraction divided by the bound fraction.

We do not believe the “double band” in the bound fraction of the a-uL3 is real; instead, an artefact that occurred either during the electrophoresis or the transfer into the membrane.

Query 5

Line 149-50: “methodologies section of the main article.” Please explain.

Response

We have deleted the phrase as it was out of context.

Reviewer 2 Report

The end step of 80S ribosome maturation is the release of the anti-association factor eIF6 (Tif6 in yeast) from the 60S. This is achieved by the coordinated action of the GTPase Elongation Factor-like1 (EFL1) and the Shwachman-Diamond-Bodian Syndrome protein SBDS (SDO1 in yeast). These proteins also control the quality of the P-site and the GTPase center. In this manuscript, the authors investigate the impact of a mutation R1086Q in yeast Efl1 protein. This mutation is the equivalent of mutation R1095Q that is found in the human homolog EFL1 and that have been described in Shwachman-Diamond Syndrome patients. By using Isothermal Titration Calorimetry (ITC) with pure recombinant components, the authors characterized the kinetics of the enzyme and studied the energetic basis to guanine recognition. They showed that the mutant EFL1 cannot release efficiently Tif6 from the 60S. Based on their results, they propose that mutation R1086Q disrupts a long-distance network of interactions that enable conformational changes. Overall, the experimental strategy is appropriate, the conducted experiments are carefully explained and the results are properly interpreted. The authors conclude the manuscript with a complete model of the interaction between EFL1, Sdo1 and GTP and GDP with all the kinetic parameters. One major concern is the lack of comparisons with the wild-type EFL1 in most of the conducted experiments.

Specific points:

- In figure 1, the effect of the mutation on Tif6 release detected in the free fraction is not very strong, therefore this assay should be carefully quantified and repeat at least three times to have reliable statistics on the effect of R1086Q mutation. The double mutant R1086Q P151L should also be tested using this assay.

- Although thermodynamic parameters for the binding of wild-type EFL1 to guanine nucleotides are available for some experiments in the literature (Luviano et al., 2019), it is critical that the results obtained with the R1086Q and R1086Q P151L mutants are systematically compared to the Wt protein using the same experimental set-up.

Minor point:

- P6 line 253, there is a problem with the reference describing the tif6-release assay.

Author Response

Manuscript Biomolecules_1847364

Altered conformational landscape upon sensing guanine nucleotides in a disease mutant of the Elongation Factor-like 1 (EFL1) GTPase

Response to reviewer 2

Query 1

In figure 1, the effect of the mutation on Tif6 release detected in the free fraction is not very strong, therefore this assay should be carefully quantified and repeat at least three times to have reliable statistics on the effect of R1086Q mutation. The double mutant R1086Q P151L should also be tested using this assay.

Response

We quantified the signal of each band and presented the number at the bottom of each corresponding sample in Figure 1 as the signal ratio of the free fraction divided by the bound fraction. A similar experiment was presented in Tan et. al. Blood (2019) 134:277-290 with only a qualitative representation.

We agree with the reviewer it would have been advisable to repeat the release assay at least three times and perform three independent blots to have reliable statistics on the effect of R1086Q and R1086Q P151L mutations. However, we do not deem it necessary for the purpose of this manuscript as this experiment was meant as a contribution to the argument that the reason Tif6 is relocated to the cytoplasm of yeast Efl1 R1086Q cells is due to a malfunction of the protein that cannot release it and so it was worth inquiring the impairment on the protein. Not to mention that the 10 days granted by the editor to answer the reviewers’ queries will not be enough time to repeat the experiments since only growing the cells and purifying the 60S subunits take one week, plus the time needed to express and purify the three proteins, to subsequently perform the release assay and the western blot.

Query 2

Although thermodynamic parameters for the binding of wild-type EFL1 to guanine nucleotides are available for some experiments in the literature (Luviano et al., 2019), it is critical that the results obtained with the R1086Q and R1086Q P151L mutants are systematically compared to the Wt protein using the same experimental set-up.

Response

Contrasting the data presented here for the mutant Efl1 R0186Q with those previously reported for the wild-type protein in Luviano et. al. is vital to understand the alterations arising by this mutation. As such we agree with the reviewer and presenting some of the data concerning the wild-type within this manuscript would facilitated the reader rather that going back to the previous paper. In this regard, we have added data corresponding to wild-type EFL1 in the corresponding tables along the manuscript. Only that at 30 °C was added not to clog the tables with data, as already there is a large amount of information to process for the reader, and the information at 30 °C is representative of the differences between mutant and wild-type proteins and corresponds to the optimal growth temperature of yeast considering we used the yeast Efl1 orthologue.

Wild-type data were added in parenthesis next to the corresponding information of the mutant and explained at the bottom of the corresponding table. In Luviano et. al the interaction of the complex Efl1 WT·Sdo1 with the guanine nucleotides was not evaluated in the presence of magnesium ions, so in Tables 2, 3 and 4 these data are not presented.

Query 3

P6 line 253, there is a problem with the reference describing the tif6-release assay.

Response

We have corrected the cross-reference to Figure 1.

Round 2

Reviewer 1 Report

The authors have addressed all my concerns and suggestions. As such, I can now recommend the manuscript for publication